# Unique activities of two overlapping *PAX6* retinal enhancers

Kirsty Uttley[1] , Andrew S Papanastasiou[1] , Manuela Lahne[2], Jennifer M Brisbane[1] , Ryan B MacDonald[2] , Wendy A Bickmore[1] , Shipra Bhatia[1]

Enhancers play a critical role in development by precisely modulating spatial, temporal, and cell type-specific gene expression. Sequence variants in enhancers have been implicated in diseases; however, establishing the functional consequences of these variants is challenging because of a lack of understanding of precise cell types and developmental stages where the enhancers are normally active. *PAX6* is the master regulator of eye development, with a regulatory landscape containing multiple enhancers driving the expression in the eye. Whether these enhancers perform additive, redundant or distinct functions is unknown. Here, we describe the precise cell types and regulatory activity of two *PAX6* retinal enhancers, HS5 and NRE. Using a unique combination of live imaging and single-cell RNA sequencing in dual enhancer–reporter zebrafish embryos, we uncover differences in the spatiotemporal activity of these enhancers. Our results show that although overlapping, these enhancers have distinct activities in different cell types and therefore likely nonredundant functions. This work demonstrates that unique cell type-specific activities can be uncovered for apparently similar enhancers when investigated at high resolution in vivo.

## Introduction

Regulatory elements such as enhancers control the activation of target genes in precise spatial and temporal patterns, ensuring proper gene expression and the successful development of complex organisms (Long et al, 2016). The human genome contains millions of predicted enhancers, and noncoding mutations affecting enhancers can cause Mendelian disease, and contribute to complex phenotypes and drive evolutionary differences between species (Lettice et al, 2003; ENCODE Project Consortium, 2012; Maurano et al, 2012; Smemo et al, 2012; Bhatia et al, 2013; Long et al, 2020). Unravelling the functions of noncoding elements is key to understanding the regulatory rules of enhancers and the consequences of sequence variation. However, understanding the functions of enhancers remains a fundamental challenge (Meuleman et al, 2020; Jindal & Farley, 2021). In contrast to protein-coding sequences, it is not possible to predict the function of an enhancer from the sequence alone. Transcription factor-binding sites within enhancers often correspond to suboptimal binding motifs, making it hard to predict which binding sites and transcription factors are important for function (Farley et al, 2015, 2016). Another confounding problem is the cooperativity, or perhaps redundancy, of enhancers. Several studies have shown that mutation or loss of an enhancer does not necessarily cause observable, gross phenotypes in animal models (Antosova et al, 2016; Osterwalder et al, 2018; Kvon et al, 2020; Snetkova et al, 2021). This may be because of the buffering of enhancer loss by multiple elements driving similar patterns of gene expression, acting redundantly. Such overlapping enhancers can however have unique or additive effects, but the phenotypes arising from the mutation of these elements can be subtle and highly cell type-specific (Dickel et al, 2018; Long et al, 2020). It is therefore important to understand the precise functions of enhancers, particularly for elements with similar tissue-specific domains of activity.

The expression of pleiotropic developmental genes is controlled by multiple tissue-specific enhancers, and it is common for such genes to have multiple elements with apparently similar spatiotemporal activities (Kvon et al, 2021). An example is the regulatory landscape of *PAX6*, encoding a developmental transcription factor which, among other functions, is a master regulator of eye development, controlling functions ranging from the specification of the eye field to maintaining progenitor populations, and influencing the differentiation of multiple cell types (van Heyningen, 2002). The large *PAX6* regulatory landscape contains several identified enhancers active in overlapping domains of the developing lens and retina (Kammandel et al, 1999; Kleinjan et al, 2001; McBride et al, 2011; Ravi et al, 2013; Bhatia et al, 2014; Lima Cunha et al, 2019) (Fig 1A). The sequence of *PAX6* is highly conserved from flies to humans, as is its role in the specification of eye development (Halder et al, 1995; Onuma et al, 2002). Human *PAX6*-regulatory

[1]MRC Human Genetics Unit, Institute of Genetics and Cancer, The University of Edinburgh, Edinburgh, UK   [2]UCL Institute of Ophthalmology, University College London, Greater London, UK

Correspondence: shipra.bhatia@igmm.ed.ac.uk

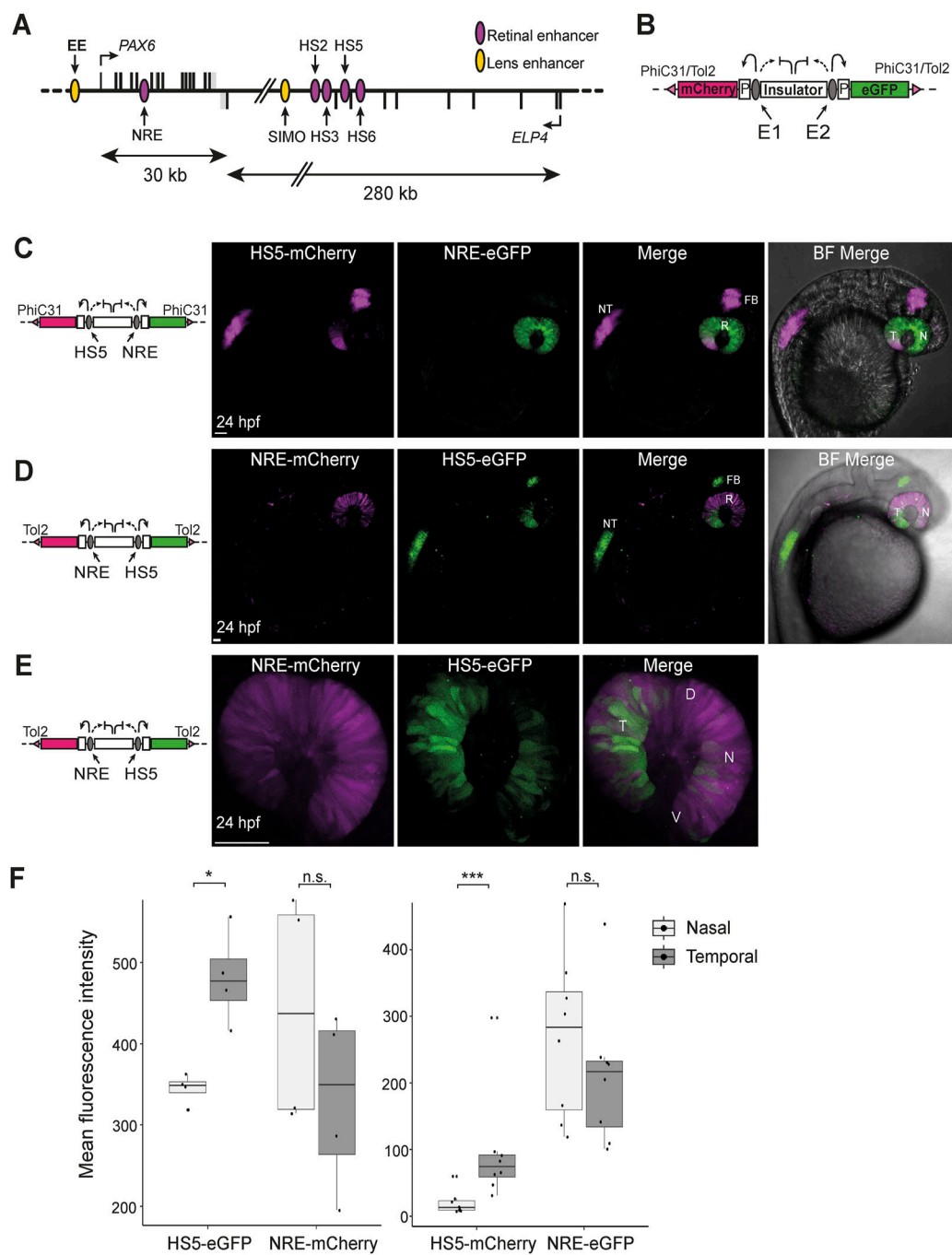

**Figure 1. Activity of HS5 and NRE in a dual enhancer–reporter system during zebrafish embryonic development.**
**(A)** Map of human *PAX6* regulatory locus showing the position of eye enhancers including HS5 and NRE (retinal, purple; lens, yellow). **(B)** Dual enhancer–reporter injection construct. mCherry and eGFP are transcribed from a minimal *gata2* promoter (P) activated by enhancer E1 or E2. An insulator based on the chicken HS4 sequence separates the enhancers. Targeted PhiC31 integration or random Tol2 integration is used to insert the dual-reporter construct into the zebrafish genome (Bhatia et al, 2021). **(C)** Live imaging of a 24-hpf NRE-eGFP/HS5-mCherry F1 embryo (10x objective). NRE (eGFP) is active throughout the retina (R). HS5 (mCherry) is active in the forebrain (FB), neural tube (NT), and in the retina where activity is highest in the temporal (T) half of the retina, compared with the nasal (N) side. **(D)** Live imaging of a 24-hpf NRE-mCherry/HS5-eGFP F1 embryo showing the activity of HS5 (eGFP) in FB, NT, and predominantly the temporal retina, towards the ventral (V) side as opposed to dorsal (D). NRE (mCherry) is active throughout the retina (10x objective); **(E, D)** as in (D), but at higher resolution (40x water immersion objective). Scale bars 50 μm.
**(F)** Quantification of mean fluorescence intensity for mCherry and eGFP in the nasal versus temporal retina at 24 hpf in NRE-mCherry/HS5-eGFP (left) and NRE-eGFP/HS5-mCherry (right) F1 embryos. The activity of HS5 (eGFP or mCherry) is significantly higher in the temporal retina. n F1 embryos imaged ≥4. Wilcoxon test results: ns, not significant; *, $P < 0.05$; ***, $P < 0.001$. Scale bars 50 μm.

elements, and their activities, also show high levels of conservation in vertebrate genomes (Williams et al, 1998; Griffin et al, 2002; Tyas et al, 2006; Bhatia et al, 2014). Mutations affecting these *PAX6* regulatory elements, ranging from large-scale genome rearrangements to single-nucleotide point mutations, can cause eye malformations such as aniridia (Lauderdale et al, 2000; Kleinjan et al, 2001; Bhatia et al, 2013). As is the case for other developmental loci, understanding the functions of individual *PAX6* enhancer elements, and whether they act in redundant, additive, or distinct ways, will help to decode noncoding mutations at this locus and further our understanding of the mechanisms of enhancer activity during development.

Here, we use a previously developed dual enhancer–reporter assay in *Danio rerio* (zebrafish) to dissect the activity of two overlapping human *PAX6* retinal enhancers, HS5 and NRE (Bhatia et al, 2021). This assay has previously been used to study the activity of human enhancers and recapitulates known patterns of activity for WT and mutant sequences (Bhatia et al, 2021). Compared with enhancer–reporter assays in cell lines, this assay preserves the tissue-specific context of enhancer activities and allows visualisation of their spatial and temporal domains of activity in live animals during development, and in-depth characterisation and comparison of two enhancers in the same embryo. Using a unique combination of live-imaging and single-cell RNA sequencing (scRNA-seq), we uncover differences in the spatial, temporal, and cell-type activities of HS5 and NRE. Our results demonstrate how distinct differences between two enhancers with overlapping activities can be revealed by high resolution in vivo analysis.

## Results

### HS5 and NRE have unique patterns of spatial and temporal activity in the developing retina

HS5 and NRE are two human *PAX6* enhancers (Fig 1A) known to be active in the developing retina, with an apparent overlap in their domains of activity (Plaza et al, 1995; Kammandel et al, 1999; McBride et al, 2011). The activity of these elements has been assessed individually in enhancer–reporter assays, using highly conserved sequences from mouse, quail, lamprey, and elephant shark (Plaza et al, 1995; Kammandel et al, 1999; McBride et al, 2011; Ravi et al, 2013, 2019). Studies on the mouse NRE sequence (also referred to as α-enhancer) have indicated that NRE is active in retinal progenitors and amacrine cell development (Marquardt et al, 2001; Kim et al, 2017; Dupacova et al, 2021). However, the precise stage and cell type-specific functions of these enhancers have not been fully characterised. To precisely define and directly compare the functional activities of HS5 and NRE, we created a dual enhancer–reporter zebrafish line using the QSTARZ system (Fig 1B) (Bhatia et al, 2021). This line contains a dual enhancer–reporter cassette, with the enhancers separated by insulators, inserted into the zebrafish genome using PhiC31 recombination at a single, known, landing pad site. In this line, NRE can activate the expression of eGFP, whereas HS5 can activate mCherry (Fig 1C). Thus, the activity of these enhancers during zebrafish development can be visualised by the expression of eGFP and mCherry. We used time-

lapse and live imaging of these NRE-eGFP/HS5-mCherry enhancer–reporter embryos to compare the spatiotemporal activities of HS5 and NRE during development. This recapitulates the known domains of activity for these elements. 24 hours post-fertilisation (hpf), NRE-eGFP is active within the retina, whereas HS5-mCherry can be visualised in the retina and the neural tube and forebrain (Fig 1C and Video 1). The activity of both of these elements in the retina peaks between 24–48 hpf, and decreases thereafter (Video 1). We confirmed this result by creating a dye-swapped dual enhancer–reporter line using random Tol2 integration on a WT background, in which NRE now activates mCherry, whereas HS5 can activate eGFP (NRE-mCherry/HS5-eGFP) (Fig 1D).

In both reporter lines, within the retina, we observed an apparent enrichment of HS5 activity specifically in the temporal portion of the retina (Fig 1C–E). This was quantified by comparison of the mean eGFP and mCherry fluorescence in the two sides of the retina. There was no significant difference in overall fluorescence intensity for NRE between the nasal and temporal sides of the retina, but HS5 activity was significantly higher in the temporal part of the retina (Fig 1F).

To investigate this further, we carried out high-resolution live imaging of multiple NRE-eGFP/HS5-mCherry enhancer–reporter embryos at 24, 48, and 72 hpf. We observed a clear difference in the spatial activity of these enhancers within the retina, with NRE broadly active throughout and HS5 highly active in the temporal portion at all three time-points (Fig 2A). Quantification of mean fluorescence intensity confirmed significantly higher HS5 (mCherry) signal in the temporal versus nasal retina from 24–72 hpf, which was not observed for NRE (eGFP) (Fig 2B). As a control, we created a dual enhancer–reporter line containing the NRE enhancer at both positions in the cassette (NRE-eGFP/NRE-mCherry) (Fig S1A). Imaging of these embryos showed a clear overlap between eGFP and mCherry expressions throughout the retina (Fig S1A). Quantification of mean fluorescence intensity at 48 and 72 hpf showed no significant difference in the temporal versus nasal signal for mCherry or eGFP (Fig S1B). However, we observed that the fluorescence intensity for NRE-mCherry at 24 hpf was significantly higher in the nasal retina (Fig S1B). This is consistent with the NRE-eGFP and NRE-mCherry measurements in the HS5/NRE reporter lines, which show a modest but not significant increase in NRE activity in the nasal versus temporal retina at 24 hpf (Figs 1F and 2B). We conclude that at 24 hpf, NRE activity is modestly higher in the nasal retina, the direct inverse of HS5 whose activity is highest in the temporal retina from 24–72 hpf. These contrasting patterns of activity led us to speculate that HS5 and NRE have distinct functions in PAX6-mediated retinal development.

### scRNA-seq reveals that HS5 and NRE are active in distinct cell types in the developing zebrafish retina

To define the precise cell types within the retina where HS5 and NRE are active, we carried out scRNA-seq on retinae from NRE-eGFP/HS5-mCherry reporter embryos. With this technique, we aimed to uncover cell type or transcriptional differences between the two enhancer-active populations. We dissected eyes from 48 hpf NRE-eGFP/HS5-mCherry embryos (F1) and used FACS to enrich for either mCherry-positive/HS5-active cells or eGFP-positive/NRE-active

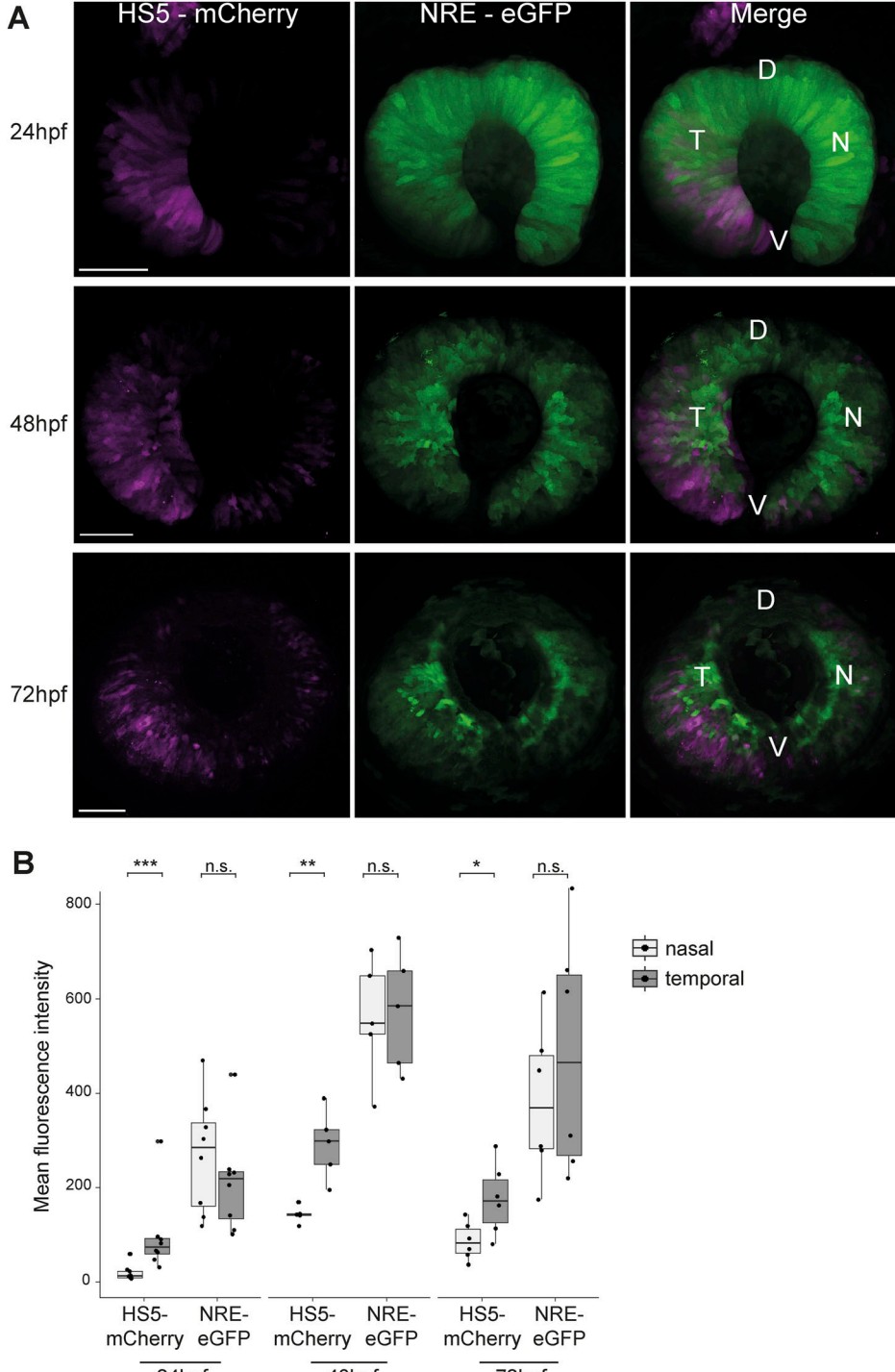

**Figure 2. HS5 and NRE are active in different zones of the developing retina.**
**(A)** Live imaging at 24, 48, and 72 hpf in the developing retina of NRE-eGFP/HS5-mCherry F1 embryos. D, dorsal; V, ventral; T, temporal; N, nasal. Scale bars 50 μm. **(B)** Quantification of mean fluorescence intensity for mCherry (HS5) and eGFP (NRE) in the nasal versus temporal retina at 24, 48, and 72 hpf. Each measurement represents one embryo. The activity of HS5 (mCherry signal) is significantly higher in the temporal retina at all time points. n F1 embryos imaged ≥6 for all time points. Wilcoxon test results: ns, not significant; *, $P < 0.05$; **, $P < 0.01$; ***, $P < 0.001$.

cells (Fig S2). Three samples for each population were processed for scRNA-seq using the 10x Genomics Chromium single-cell 3′ gene expression technology. After quality control (QC) and filtering we carried out K-nearest neighbour analysis followed by Louvain clustering on 6,288 cells, revealing 13 distinct clusters (Fig 3A). We used enriched marker genes to annotate each cluster, using gene expression data from ZFIN and the literature for known retinal cell

types (Sprague et al, 2008) (Figs 3B and C and S3 and Table S1). A large proportion of the cells were classified as progenitors or stem cells, marked by high expression of *pcna* (proliferating cell nuclear antigen). As expected, these clusters were also characterised as actively proliferating (S and G2/M), in contrast to the more differentiated cell-type clusters (Fig S4A). The stem cell cluster has high expression of genes such as *fbl* and *fabp11a* (Watanabe-Susaki

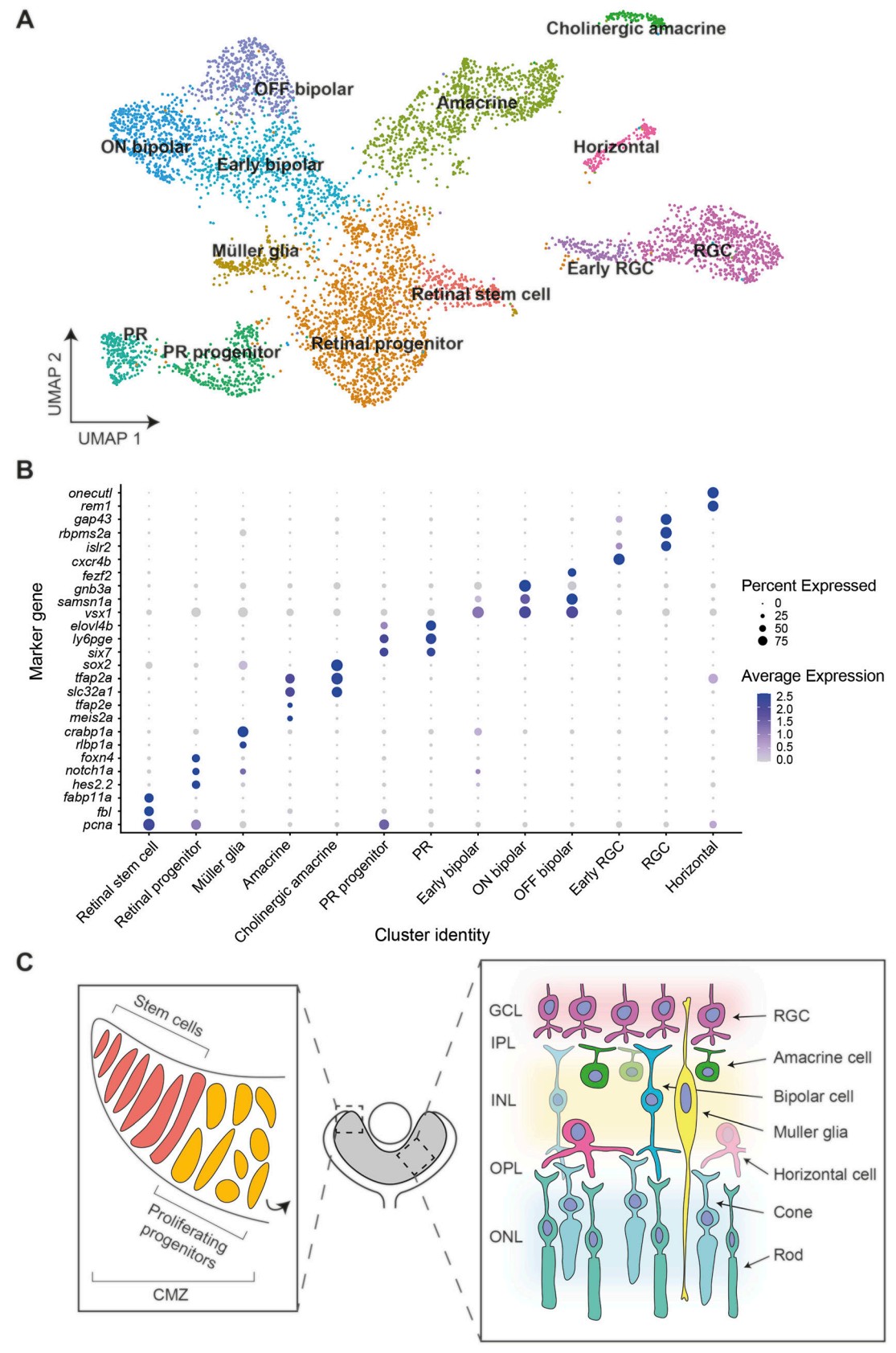

et al, 2014), whereas the general retinal progenitor cluster is marked by high expression of the Notch-signalling genes *notch1a* and *hes2.2*, and the progenitor marker *foxn4* (Li et al, 2004). In addition to these cell types, we identified clusters of amacrine, horizontal, bipolar, Müller glia, photoreceptor (PR), and retinal ganglion cells (RGCs). We were also able to annotate subtypes such as a cluster of OFF bipolar cells expressing *fezf2* (Suzuki-Kerr et al, 2018), and cholinergic amacrine cells, distinguished from the general amacrine cell cluster by the expression of *sox2* (Whitney et al, 2014) (Fig 3A and B).

We also used the zebrafish single-cell transcriptomic atlas to assign cell-type identities in our dataset (Farnsworth et al, 2019). Most of the cells were assigned to "retinal differentiating" and "retinal progenitor" identities (Fig S4B). The "retinal progenitor" identity appears to specifically mark cells from the retinal stem cell cluster (Fig S4B). Comparing the assigned atlas identities with our cell-type annotations in more detail proved informative. For example, the atlas identity RetNeuron25 appears to be specific to amacrine cells, whereas RetDiff25e appears to mark RGCs (Fig S4C).

To confidently identify the cell types where the HS5 and NRE enhancers are active, we looked for the presence of mCherry and eGFP reads, respectively. We observed the highest expression of eGFP reads within both the amacrine and stem cell clusters, and the highest expression of mCherry reads within the Müller glia and retinal progenitor/stem cell clusters (Fig 4A–C). Importantly, all of these cell-type clusters also have high expression of zebrafish *pax6a* and *pax6b*, making them conceivable candidates for *PAX6* enhancer-active cell types (Fig S5A–C). Although we used FACS to enrich for eGFP- and mCherry-positive cells, most cells within our dataset do not express eGFP or mCherry. We expect the reason for this to be twofold. Firstly, "dropout" of reads in scRNA-seq data is common, particularly if the gene is lowly expressed and there is low mRNA in individual cells, which we expect in our samples (Kharchenko et al, 2014). It is likely that "dropout" of eGFP and mCherry reads has occurred in clusters where we see highest expression of these genes. Secondly, false-positive selection can occur during FACS, particularly where there is autofluorescence within the sample, which is true for zebrafish eyes (Shi et al, 2009).

Because of the lower than expected number of eGFP and mCherry reads in our scRNA-seq dataset, we employed differential abundance analysis to validate the mCherry-enriched (HS5) and eGFP-enriched (NRE) clusters. We used two statistical methods of differential abundance analysis (DAseq and MiloR) to calculate the relative abundance of cells from eGFP- or mCherry-enriched samples within regions of our Seurat clusters (Zhao et al, 2021; Dann et al, 2022). Both of these methods showed regions of

enrichment for cells from mCherry-enriched samples within retinal progenitor and Müller glia clusters, and enrichment for cells from eGFP-enriched samples within the amacrine and stem cell clusters when visualised on a UMAP plot (Figs 4D and S6A) and in the DAseq scores per cluster (Fig S6B).

We also applied topic modelling (Dey et al, 2017) to our dataset to simultaneously identify gene topics (corresponding to sets of coexpressed genes) and cell-topic weights (which quantify the proportion of a cell's transcriptome described by a given gene topic). We then used a two-sided Pearson's product–moment correlation test to find topics correlated with the expression of mCherry and eGFP (Table S2). We find that topic 1 from this analysis was most highly correlated with mCherry expression, and contributes highly to cells from Müller glia and progenitor clusters (Fig S6C). Inspecting the genes of this topic reveals that it is enriched for notch-signalling genes and Müller glia markers, again giving confidence that mCherry expression/HS5-activity appears enriched in cells of the progenitor and Müller glia clusters (Table S3). The topic most highly correlated with the expression of eGFP (topic 5) contributes highly to cells in the amacrine clusters, and again is enriched for genes characteristic of amacrine cells, including *tfap2a* and *slc32a1* (Fig S6D and Table S4). Thus, the topics most highly correlated with eGFP and mCherry expression correspond to topics that are characteristic of the cell-type clusters we find using differential abundance analysis. The agreement between the cluster-specific enrichment of eGFP and mCherry reads with the differential abundance analysis and topic modelling suggests that our coupling of the QSTARZ assay with scRNA-seq analysis is a powerful way of identifying the precise cell types within a tissue where enhancers are active.

## HS5 and NRE-active cell types are confirmed by immunofluorescence

Our scRNA-seq dataset indicates that NRE is primarily active in amacrine and retinal stem cells, whereas HS5 appears to be active in proliferating progenitors and Müller glia (Fig 4). This is in agreement with our live imaging data detecting NRE-eGFP signal in the distal tip of the ciliary marginal zone (CMZ) and in cells of the inner nuclear layer (INL), and HS5-mCherry signal in the more proximal region of the CMZ and in cells of the INL, at 48 hpf (Fig 5A). To confirm the identity of these enhancer-active cell types, we carried out immunofluorescence on eye sections from NRE-eGFP/HS5-mCherry embryos. We stained for PCNA—a marker of progenitor and stem cells in the CMZ, HuC/D (encoded by *elavl3/4*)—a marker for RGCs in the ganglion cell layer and amacrine cells in the INL, glutamine synthetase (GS, encoded by *glula*)—a marker

---

**Figure 3. Single-cell RNA sequencing of NRE-eGFP/HS5-mCherry retinal cells at 48 hpf.**
**(A)** Cells from six sequenced libraries of NRE-eGFP/HS5-mCherry F1 embryos at 48 hpf, merged into one sample, and visualised on a Uniform Manifold Approximation and Projection plot created by Louvain clustering using Seurat (Butler et al, 2018). Retinal cell types are manually annotated to clusters depending on marker gene expression. **(B)** Dot plot showing average expression level and percentage of cells expressing key marker genes used to annotate cell-type clusters. **(C)** Schematic of cell types in the zebrafish retina. In the ciliary marginal zone, retinal stem cells undergo asymmetrical division to give rise to a rapidly proliferating pool of retinal progenitor cells (left), which divide and differentiate to form cells present in the retinal layers (right) (Richardson et al, 2017; Wan et al, 2016). **(A)** Colours correspond to the annotated cluster cell types in (A). RGC, retinal ganglion cell; GCL, ganglion cell layer; IPL, inner plexiform layer; INL, inner nuclear layer; OPL, outer plexiform layer; ONL, outer nuclear layer.

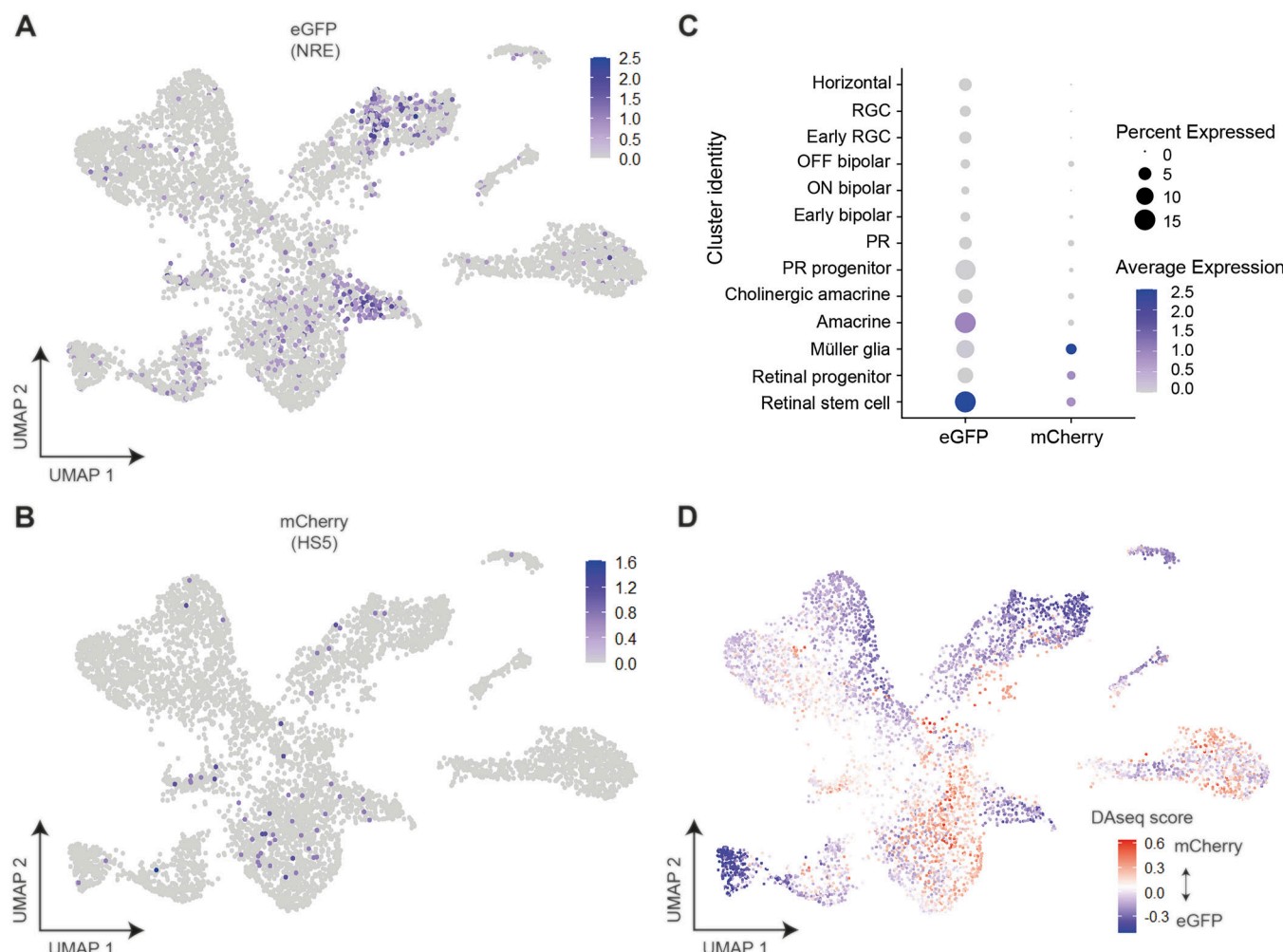

**Figure 4. Assigning the identity of HS5 and NRE-active cells using the expression of eGFP and mCherry, and differential abundance analysis in cell-type clusters.**
**(A, B)** Expression of eGFP and (B) mCherry in single cells visualised on UMAP plots (clustered as in Fig 3A). eGFP expression is enriched in amacrine and retinal stem cell clusters. **(C)** Dot plot showing average expression and percentage of cells expressing mCherry (HS5-active) and eGFP (NRE-active) in cell-type clusters. Enrichment is seen for eGFP expression in amacrine and retinal stem cell clusters, and for mCherry in Müller glia and retinal progenitor/stem cell clusters. **(D)** DAseq (Zhao et al, 2021) differential abundance analysis comparing the relative prevalence of cells from mCherry-enriched or eGFP-enriched samples. Cells are coloured by DAseq score and displayed on a UMAP plot. A score is calculated for each cell based on the abundance of cells from both populations in the cell's neighbourhood. Positive (red) scores indicate an abundance of cells from mCherry-enriched samples; negative (blue) scores indicate an abundance of cells from eGFP-enriched samples.

for Müller glia (Ekström & Johansson, 2003; Thummel et al, 2008; Fischer et al, 2013). Our scRNA-seq data confirm the expression of these genes within these specific cell types in our samples (Fig 5B).

We detect strong expression of eGFP in PCNA+ cells of the CMZ, particularly in the stem-cell region in the distal tip (Fig 5C, arrowhead). We also observed mCherry expression in PCNA+ cells; however, in contrast to eGFP, this is seen in the more proximal region of the CMZ, where the rapidly proliferating progenitors are located (Fig 5C, arrow). Staining for HuC/D and parvalbumin (pv), we observed co-staining with eGFP in amacrine cells of the INL (Figs 5D and S7A and B arrowheads). We also detected mCherry expression in a subset of GS+ Müller glia at 72 hpf (Fig 5E, arrowheads). These results confirm our findings that HS5 and NRE are active in distinct cell types of the developing retina, with NRE active in amacrine and

retinal stem cells, and HS5 active in proliferating progenitors and Müller glia.

## Discussion

Several issues challenge our understanding of the precise role of enhancers in developmental and disease processes, including relating the sequence-specific grammar of these elements to their function, defining their exact domains of action, and disentangling the additive, redundant or distinct functions of apparently similar enhancers. A few enhancers, such as the famous ZRS limb enhancer of *SHH*, have been so thoroughly characterised that they can now be manipulated at the single base-pair level to achieve specific

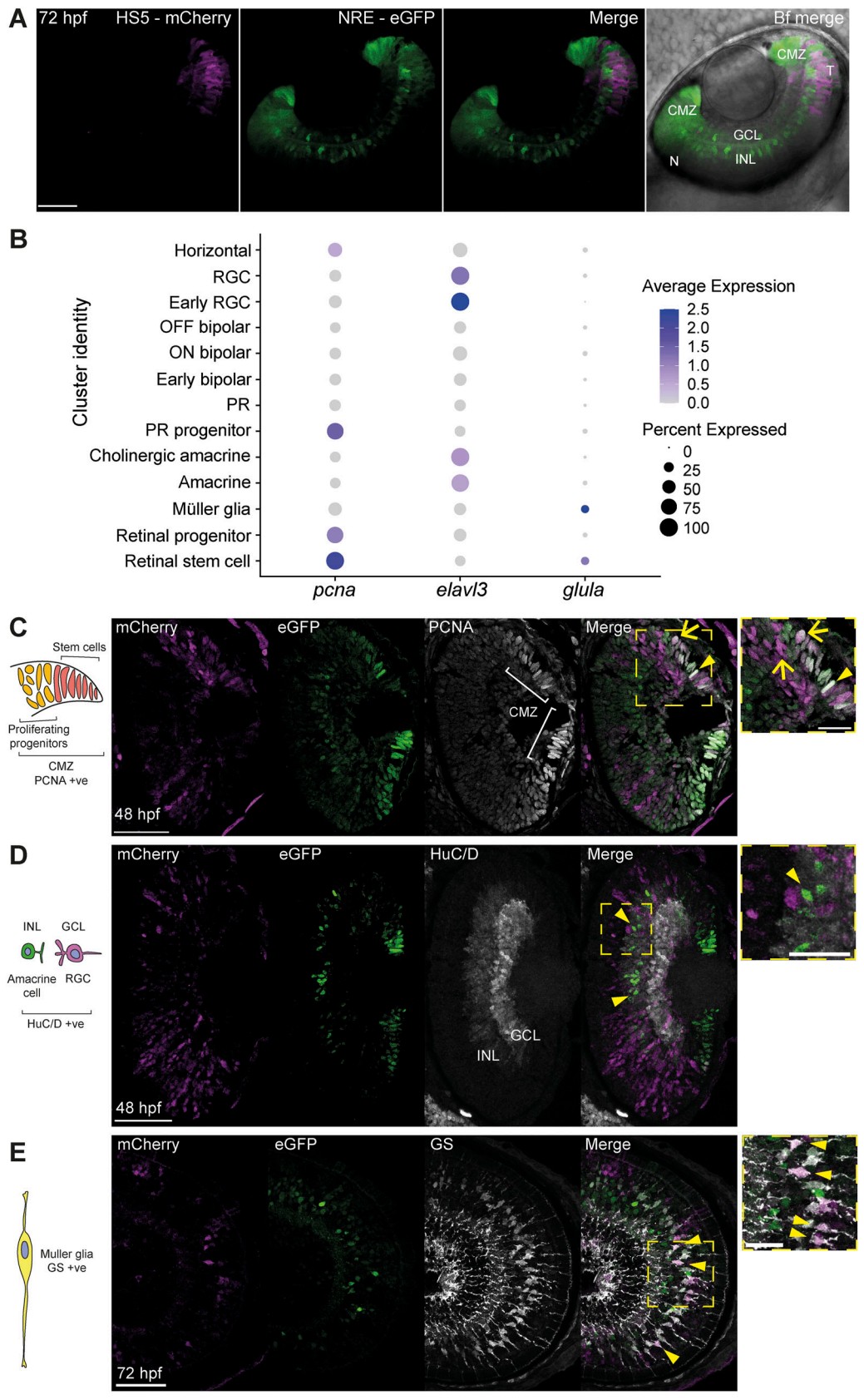

phenotypes (Lettice et al, 2003; Kvon et al, 2020; Lim et al, 2022 Preprint). However, our understanding of the other hundreds of thousands of predicted enhancers in the human genome lags far behind this, and for the majority, it remains challenging to identify even the target gene with confidence. Effective and efficient in vivo analysis pipelines for characterisation of regulatory elements are needed to address this knowledge gap. In this study, we have described an in vivo method for generating high-resolution data on the precise activities of developmental enhancers. Using a combination of live imaging and scRNA-seq in embryos derived from dual enhancer–reporter transgenic zebrafish lines, we clearly demonstrate distinct patterns of spatiotemporal and cell type-specific activity during retinal development for two retinal enhancers (HS5 and NRE) from the PAX6 regulatory region (Fig 6).

It has been shown that NRE is active in a retinal progenitor population during mouse embryonic development (Marquardt et al, 2001). In this study, we have shown that NRE is specifically active in retinal stem cells of the CMZ, as can be seen by scRNA-seq (Figs 3 and 4) and high-resolution live-imaging and immunofluorescence (Fig 5). NRE has also been reported as active in non-cholinergic amacrine cells in postnatal mouse eyes, and necessary for their development (Kim et al, 2017). It is also observed as an active enhancer in GABAergic amacrine cells in the scATAC-sequencing data generated from adult human retinae (Wang et al, 2022). This mirrors the known functions of PAX6, which is essential for the generation of amacrine cells (Remez et al, 2017). Here, we show that as early as 48 hpf in the zebrafish (~E14.5 in mouse), NRE is seen to be strongly active in differentiating amacrine cells by immunofluorescence imaging (Fig 5), in specifically the non-cholinergic population, as is shown by scRNA-seq (Figs 3 and 4). It is likely that the reduced amacrine cell phenotype reported by Kim et al (2017) upon NRE deletion is because of the loss of NRE activity at these early time points affecting PAX6 expression. The dual activity of NRE in retinal stem cells and differentiating amacrine cells reflects the multifarious functions of PAX6, which is necessary both for promoting the proliferation and potency of retinal progenitors, and directing cell-cycle exit and differentiation of cell types including amacrine cells (Farhy et al, 2013; Remez et al, 2017).

Previously, little was known about the precise function of HS5 in the developing retina (McBride et al, 2011). Here, we show that HS5 is active in the rapidly proliferating progenitor population of the developing retina, and Müller glia (Figs 3, 4, and 5). HS5 is also detected as an active enhancer in Müller glia cells in scATAC-sequencing data from adult human retinal samples (Wang et al, 2022). Transcriptionally, Müller glial cells share similarities with retinal progenitor cells, in that they are specialised glial cells

with progenitor potential (Jadhav et al, 2009). Again, it is known that PAX6 expression is detected in both of these cell types, and is necessary for the maintenance of progenitors and generation of Müller glia (Marquardt et al, 2001; Joly et al, 2011). In zebrafish, Müller glial cells are capable of undergoing transcriptional reprogramming to produce retinal progenitor cells after an acute injury to the retina (Goldman, 2014). This property has also been shown for human Müller glia in vitro and in rodent transplants (Singhal et al, 2012). During development, each retinal progenitor cell is capable of forming several neural retina cells and a single Müller glia (Rulands et al, 2018). Whether HS5 is active in Müller glia at later stages or if the activity of HS5 in these cells is linked to Müller glia development and HS5 activity in upstream retinal progenitors is unclear.

HS5 and NRE also showed distinct differences in their spatiotemporal patterns of activity. The activity of HS5 in the temporal retina is notable, as in zebrafish, this region is a fovea-like region of high acuity named the "area temporalis," characterised by specialisation and increased density of cell types (Schmitt & Dowling, 1999; Yoshimatsu et al, 2020). It is unclear if HS5 may also show a similar pattern of activity in the human fovea, and whether this may be functionally relevant to fovea development. Of note, Müller glial cells play a key role in the fovea, where they are one of only two cell types (the other being cone photoreceptors), and provide essential support to the structural integrity of this region (Bringmann et al, 2018; Delaunay et al, 2020). The activity of HS5 in Müller glia and the "area temporalis" could therefore be linked. Whereas HS5 shows greatest activity in the temporal zebrafish retina, the fluorophore expression driven by this enhancer is still measurable at lower levels in the nasal retina (Fig 2). It is therefore unclear as to what extent this enhancer drives PAX6 expression in cell types in this region. The previous study identifying HS5 in a mouse lacZ reporter assay did not report any specific spatial activities for this enhancer within the developing eye, and the resolution of this assay would be poorly suited to do this (McBride et al, 2011).

A limitation of this study, and enhancer–reporter assays in general, is the fact that the enhancers are tested outside of their native genomic context. An advantage of this dual enhancer–reporter assay in zebrafish, however, is that the developmental context of enhancer activity is preserved, and this can be easily followed in live embryos, particularly in developing eyes. The transgenic lines also serve as a valuable resource for isolating enhancer-active cell populations, whose precise identity can be determined by using scRNA sequencing. This generates an enhancer-centric view of the cell types and stages of development where the expression of the target gene is potentially regulated by

**Figure 5. Immunofluorescence identifies enhancer-active cell types.**
**(A)** Coronal-orientation image of a NRE-eGFP/HS5-mCherry embryo at 48 hpf showing the activity of NRE (eGFP) in the distal CMZ (stem cell niche) and cells of the INL, and HS5 (mCherry) activity in the temporal retina, in the proximal CMZ, and cells of the INL. **(B)** Dot plot showing average expression and percentage of cells expressing pcna, elavl3 (encoding HuC/D), and glula (encoding glutamine synthetase [GS]) in cell type clusters. **(C)** Immunofluorescence for PCNA, mCherry, and eGFP on a coronal eye section from an NRE-eGFP/HS5-mCherry F1 embryo at 48 hpf. PCNA is a marker for progenitors and stem cells in the CMZ. An arrow indicates an mCherry/PCNA-positive cell. An arrowhead indicates a eGFP/PCNA-positive cell. **(D)** As in (C), but using an antibody detecting HuC/D (elavl3/4). HuC/D is a marker for RGCs in the GCL and amacrine cells in the INL. Arrowheads indicate eGFP/HuC/D-positive cells. **(E)** As in (C) but using an antibody detecting GS, on an embryo at 72 hpf (sagittal section). GS is a marker for Müller glia. Arrowheads indicate mCherry/GS-positive cells. Scale bars 50 μm, 20 μm in zoom.

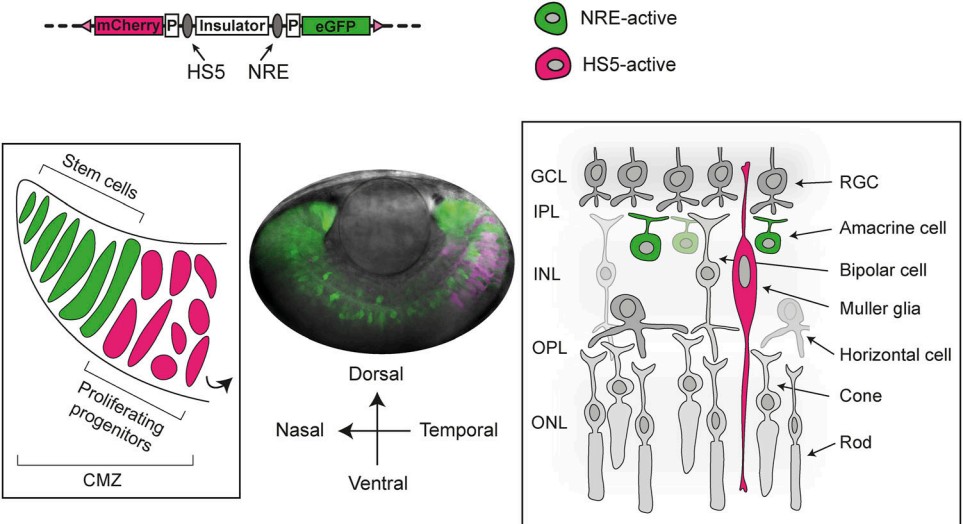

**Figure 6. Investigating the distinct functions of overlapping *PAX6* retinal enhancers in a zebrafish dual enhancer–reporter assay.**
*PAX6* retinal enhancers HS5 and NRE have distinct spatiotemporal and cell type-specific functions in a dual enhancer–reporter system in zebrafish embryonic development. NRE is active throughout the developing retina, and is localised to stem cells of the CMZ and amacrine cells of the INL (green). HS5 is active mainly in the temporal region of the retina, in proliferating progenitors, and Müller glia (magenta).

this sequence. Given the small size of the enhancer-active cell populations, this resolution cannot be achieved in scATAC-sequencing experiments, unless they are coupled with expression studies (Wang et al, 2022). As such, zebrafish are an ideal model to study the activity of the human *PAX6* retinal enhancers tested here. Zebrafish are already a well-established model for studying ocular genetics, as eyes and eye development are highly conserved between zebrafish and humans, including morphology, cell types, protein markers, and gene expression (Richardson et al, 2017; Angueyra & Kindt, 2018; Vöcking & Famulski, 2023). Species-specific differences may limit the interpretation of some results, which we have discussed.

To the best of our knowledge, ours is the first report where an in vivo system has been used to isolate, characterise, and compare the precise cell type-specific activities of developmental enhancers. We have shown that two overlapping *PAX6* retinal enhancers have distinct spatial, temporal, and cell type-specific activities, and as such, it is highly likely that they are nonredundant with differing functions in human retinal development. Based on our results, we suggest that other enhancers with apparently overlapping domains of activity based on lower resolution assays may have similar distinct functions that require proper dissection at high spatial and temporal resolution. Indeed, there are several other retinal enhancers at the *PAX6* locus that have yet to be characterised, and which may also have unique or redundant functions. The comprehensive framework described here to evaluate HS5 and NRE provides a systematic pipeline to investigate the activities of enhancers, and the consequences of enhancer mutations. The scRNA-seq dataset of developing zebrafish retinal cell types, in which different enhancers are active, sheds light on gene-regulatory networks during retinal development, and can be utilised as a valuable resource in future studies. The detailed description of the cell types and stages in development where the enhancers are active would inform analysis of functional studies deleting these sequences or assessment of phenotypes of patients with variants in these enhancers.

# Materials and Methods

### Generation of dual enhancer–reporter constructs

Dual enhancer–reporter constructs were generated using Gateway cloning (Invitrogen) as described in Bhatia et al (2021). Destination vectors containing the gata2-eGFP and gata2-mCherry gene units were synthesised by GeneArt. These contain R3/R4 Gateway recombination sites for insertion of enhancer sequences, and the cassette is flanked by either PhiC31 or Tol2 recombination sites for genome integration. The sequences of HS5 and NRE were PCR-amplified from human genomic DNA using Phusion high-fidelity polymerase (NEB). Hg38 genome coordinates and sequences of primers used, containing overhang Gateway recombination sequences, are in Tables S5 and S6. Purified PCR products were cloned into Gateway pDONR entry vectors (pP4P1r or pP2rP3) using BP clonase. Plasmids were sequenced using the original enhancer primers to verify integration. The insulator construct was previously created by cloning into a pDONR221 vector (Bhatia et al, 2021). The insulator construct used in this study contains 2.5 copies of the chicken HS4 sequence. Final dual enhancer–reporter constructs were created using LR clonase in a multi-way Gateway reaction to combine the two enhancer sequences and the insulator into the destination vector. The NRE-eGFP/HS5-mCherry and NRE-eGFP/NRE-mCherry constructs were created using the PhiC31 destination vector, and the swap construct NRE-mCherry/HS5-eGFP was created using the Tol2 destination vector.

### Generation of transgenic zebrafish lines

Transgenic zebrafish lines were created as described in Bhatia et al (2021). For the PhiC31 constructs, embryos were obtained from landing-line adults and injected at the one-cell stage. The loss of reporter gene expression from the landing pad (described in Bhatia et al [2021]) and simultaneous gain of expression driven by HS5 and NRE in the dual enhancer–reporter construct injected were used to

assess successful integration. For the Tol2 construct, embryos were obtained from WT adults (strain AB) and injected at the one-cell stage. F0s were screened for mosaic expression of eGFP and mCherry (and loss of landing-line fluorescence expression when using PhiC31) and raised to adulthood. Sexually mature F0s were crossed with WT adults and F1s screened for fluorescence using a Leica M165FC fluorescence stereo microscope. Fluorescent F1s were used for imaging and scRNA-seq. At least two F0 founders per construct were used to generate F1s.

## Zebrafish husbandry

Adult zebrafish were maintained according to standard protocols (Sprague et al, 2008). Embryos were raised at 28.5°C and staged by hpf and morphological criteria (Kimmel et al, 1995). All zebrafish work was carried out under a UK Home Office licence under the Animals (Scientific Procedures) Act, 1986.

## Live and time-lapse imaging

Before imaging, all embryos were treated with 0.003% 1-phenyl2-thio-urea (PTU) from 12 hpf to prevent pigmentation from developing. For live imaging, embryos at the correct stage were anaesthetised with Tricaine (20–30 mg/liter) and mounted in 1% low-melting point agarose in a glass-bottom dish (P06-1.5H-N; Cellvis). Embryo media were added to the dish to prevent drying out. Embryos were imaged using a Nikon A1R (scanning) confocal microscope, using a 10x objective or 40x water immersion objective. Data were acquired using NIS Elements AR software (Nikon Instruments Europe). Images were taken as a Z stack (1-$\mu$m step size). Imaging was repeated for several F1s per line (see figure legends). For time-lapse imaging, embryos were mounted as described, and a portion of the low-melting point agarose surrounding the embryo head/body was carefully cut away using a microsurgical knife (World Precision Instruments). This left only the tail of the embryo embedded in agarose, allowing unimpeded development throughout the imaging time-course. Tricaine (20–30 mg/liter) and PTU (0.003%) were added to the embryo media covering the embryos. Time-lapse imaging was carried out using an Andor Dragonfly (spinning disk) confocal (Andor Technologies), using a 10x objective. Data were collected in spinning disk 40-$\mu$m pinhole mode on the iXon 888 EMCCD camera using Andor Fusion acquisition software. Embryos were maintained at 28.5°C using an Okolab bold line stage top incubator chamber (Okolab S.R.L). Images were taken as a Z stack (1 $\mu$m step size) every 60 min for ~48 h.

## Immunostaining and imaging

NRE-eGFP/HS5-mCherry embryos were dechorionated and fixed in 4% PFA overnight at 4°C, then stored in 30% sucrose. Embryos were embedded in optimal cutting temperature compound, and flash-frozen at −80°C. Optimal cutting temperature blocks were cryosectioned and samples dried onto SuperFrost Plus Adhesion slides (Epredia). Samples were then rehydrated in PBS for 5 min at RT, followed by antigen retrieval in 10 mM sodium citrate (pH 6), heated for 20 min in a rice cooker. Once returned to RT, the samples were washed three times in PBS 0.1%

Triton X-100 (PBST) for 5 min at RT. Samples were blocked for 1 h at RT in 1% BSA, 10% goat serum in PBST, and then incubated with a primary antibody in a blocking solution overnight at 4°C. All samples were incubated with chicken anti-GFP (GTX13970; Gene Tex) (1:1,000), rabbit anti-mCherry (26765-1-AP; Proteintech) (1:1,000), and either mouse anti-PCNA (P8825; Sigma-Aldrich) (1:200), mouse anti-HuC/D (A21271; Invitrogen) (1:200), mouse anti-parvalbumin (P3088; Sigma-Aldrich) (1:200) or mouse anti-GS (66323-1-Ig; Proteintech) (1:300). After incubation slides were washed three times in PBS for 20 min at RT, followed by incubation with secondary antibody for 2 h at RT. The following Alexa Fluor-conjugated secondary antibodies were used at 1:1,000 dilution—goat anti-mouse Alexa Fluor 647 (A-21235; Invitrogen), goat anti-chicken Alexa Fluor 488 (A-11039; Invitrogen), goat anti-rabbit Alexa Fluor 546 (A-11035; Invitrogen). The samples were then washed three times in PBS for 20 min at RT, with 1 $\mu$g/ml of DAPI added to the last wash. Slides were mounted with a coverslip using mounting medium (Ab104139; Abcam), then stored in the dark at 4°C. Slides were imaged on a Zeiss LSM 900 scanning confocal using a 40x water immersion objective in z-stack acquisition mode (0.5-$\mu$m step size).

## Image analysis

All images were processed using FIJI and are displayed in figures as maximum intensity projections (Schindelin et al, 2012). eGFP and mCherry mean fluorescence intensity measurements were taken using FIJI in the temporal and nasal portions of the retina only. In R, ggplot2 and ggpubr packages were used to plot these measurements and compare means between groups using a Wilcoxon test (Wickham, 2009).

## Dissociation and sorting of zebrafish embryonic retinae

F1 NRE-eGFP/HS5-mCherry embryos, and WT embryos, were collected and treated with PTU as described. At 48 hpf, embryos were anaesthetised with Tricaine (20–30 mg/liter) and placed into Danieau's solution (Sprague et al, 2008). Eyes were dissected from ~100–150 embryos using fine forceps (#5SF; Dumont), and immediately placed into Danieau's solution on ice. Samples were centrifuged at 300$g$ for 1 min at 4°C, and then washed with Danieau's solution. The washing step was carried out three times with Danieau's solution, and once with FACSmax (Amsbio). In a final 500-$\mu$l FACSmax, the samples were passed through a 35-$\mu$m cell strainer to obtain single-cell suspension (on ice). Samples were sorted for mCherry and eGFP fluorescence using a FACS Aria II (BD) or CytoFLEX SRT (Beckman Coulter) machine. Forward and side scatter sorting was used to select single cells from clumps and debris, and DAPI staining was used to exclude dead cells. A WT sample was used as a negative control for eGFP and mCherry fluorescence to set the gates for sorting. Cells single positive for eGFP were selected for the eGFP samples. Because of the smaller population of mCherry-positive cells, the yield of mCherry samples was increased by sorting cells single positive for mCherry or double positive.

## scRNA-seq

After FACS, ~10,000 cells/sample were processed using the 10x Genomics Chromium single-cell 3′ gene expression technology

(v3.1), according to the manufacturer's instructions (Zheng et al, 2017). Sequencing was performed on the NextSeq 2000 platform (Illumina Inc.) using the NextSeq 1000/2000 P3 Reagents (100 cycles) v3 Kit. Sequencing data were processed using Cell Ranger (10x Genomics, v6.1.2). Cellranger mkfastq was used to create FASTQ files from raw sequencing data, followed by cellranger count to perform alignment, filtering, barcode counting, and UMI counting. A custom reference genome was created for alignment using cellranger mkref, combining the *Danio rerio* GRCz11 genome assembly with manually annotated eGFP and mCherry sequences.

### Computational analysis of scRNA-seq data

#### Cell calling and QC
Taking the raw (unfiltered) output from cellranger count, we used the emptyDrops function from DropletUtils to filter-out empty droplets/barcodes not corresponding to cells (Lun et al, 2019). Mitochondrial and ribosomal genes were excluded from the emptyDrops analysis to improve the filtering of droplets containing ambient RNA or cell fragments. The scater package was used to filter cells based on the QC metrics of library size, detected genes, and mitochondrial reads (McCarthy et al, 2017). Cells with detected genes ≥500, library size ≥800, and mitochondrial reads ≤10% were retained. Within the processed dataset, mean reads per cell = 11,239, and median genes per cell = 1,554.

#### Reference mapping and filtering
The SingleR package was used to annotate cell types based on mapping to the zebrafish single-cell transcriptome atlas (Aran et al, 2019; Farnsworth et al, 2019). Expression matrix and cell annotation data were downloaded from the UCSC cell browser (http://zebrafish-dev.cells.ucsc.edu); only the 2 dpf data were used for mapping. Erroneously sorted cells of non-retinal identity (for example pigmented cell types such as melanocytes with high autofluorescence) were filtered out at this stage. This was carried out to improve the resolution of clustering for retinal cell types.

#### Clustering and cell-type annotation
Seurat (v4) was used for clustering and further analysis for a total of 6,288 cells (Butler et al, 2018). SCTransform was used to perform log normalisation, scaling, and highly variable gene detection on a dataset consisting of the six samples merged into one. Standard SCTransform options were used, with regression of mitochondrial expression and cell-cycle stage using "vars.to.regress." We performed principle component analysis on the normalized counts matrix restricted to highly variable genes, using Seurat's Run principle component analysis function with number of PCs = 50. To enable integration of the samples, we then used Harmony to generate PCs corrected for batch effects between libraries (Korsunsky et al, 2019). The Harmony PCs were then used to perform K-nearest neighbour analysis (k = 20) and Louvain clustering using Seurat (15 dimensions and resolution 0.6). Clusters were annotated as retinal cell types based on the highest expressed marker genes, and other known genes for each cell type, using information from the literature and ZFIN (Sprague et al, 2008). Cell cycle scoring was performed using the Seurat CellCycleScoring function, using zebrafish genes homologous to the "s.features" and "g2m.features" genes provided by Seurat.

#### *Differential abundance analysis and topic modelling*
DAseq and MiloR were used to perform differential abundance analysis between eGFP and mCherry-enriched samples using the standard, suggested parameters (Zhao et al, 2021; Dann et al, 2022). UMAP embeddings from Seurat were used as the graphing inputs. Topic modelling was carried out using fastTopics, using k = 8 number of topics (Dey et al, 2017). Non-normalised counts were used as the input, as is standard for topic modelling. Correlation between fluorophore expression and topic scores were calculated using a two-sided Pearson's product–moment correlation using the cor.test function from R stats (R Core Team, 2021).

## Data Availability

The raw and processed scRNA-seq data generated in this study are available from the Gene Expression Omnibus (GEO) under the accession number GSE240575.

### Ethics statement

All zebrafish experiments were approved by the University of Edinburgh Ethical Committee and performed under UK Home Office licence PPL PA3527EC3.

## Supplementary Information

## Acknowledgements

We thank A Brombin and J Travnickova for their guidance and advice on scRNA-seq, and helping with eye dissections alongside G Alston, B Bartlett, L Gomez Acuna, A Gonzalez Estevez, and K Purshouse. We thank E Freyer and S Campbell of the IGC FACS facility, and R Clark of the Edinburgh Clinical Research Facility for processing 10x samples, and the MRC HGU Zebrafish facility and IGC Advanced Imaging Resource for support. Finally, we thank G Alston and E Friman for comments on the manuscript. K Uttley is funded by a PhD studentship from the UK Medical Research Council. AS Papanastasiou is a cross-disciplinary postdoctoral fellow supported by funding from the University of Edinburgh and Medical Research Council (MC_UU_00009/2). M Lahne is supported by a Moorfields Eye Charity Grant (GR001388). JM Brisbane is funded by a PhD studentship from the UK Medical Research Council. RB MacDonald is funded by a BBSRC David Phillips Fellowship (BB/S010386/1). WA Bickmore is supported by Medical Research Council University Unit Grant MC_UU_00007/2. S Bhatia is funded by a personal fellowship from the Royal Society of Edinburgh/Caledonian Research fund (RSE/CRF personal research fellowship 2014).S Bhatia and WA Bickmore are supported by a project grant from BBSRC (BB/T010509/1).

### Author Contributions

K Uttley: data curation, formal analysis, investigation, visualization, methodology, and writing—original draft.
AS Papanastasiou: data curation and formal analysis.
M Lahne: formal analysis.

JM Brisbane: formal analysis.
RB MacDonald: data curation and methodology.
WA Bickmore: writing—review and editing.
S Bhatia: conceptualization, resources, data curation, formal analysis, supervision, funding acquisition, investigation, methodology, project administration, and writing—review and editing.

## Conflict of Interest Statement

The authors declare that they have no conflict of interest.

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
